# PK Modeling of L-4-Boronophenylalanine and Development of Bayesian Predictive Platform for L-4-Boronophenylalanine PKs for Boron Neutron Capture Therapy

**DOI:** 10.3390/ph17030301

**Published:** 2024-02-26

**Authors:** Woohyoung Kim, Ji Yeong Won, Jungyu Yi, Seung Chan Choi, Sang Min Lee, Kyungran Mun, Hyeong-Seok Lim

**Affiliations:** 1Clinical Development, Dawonmedax Co., Ltd., Seoul 06735, Republic of Korea; nmwhkim@gmail.com (W.K.);; 2Treatment Planning System, Dawonmedax Co., Ltd., Seoul 06735, Republic of Korea; 3Department of Clinical Pharmacology and Therapeutics, Asan Medical Center, University of Ulsan College of Medicine, Pungnap-2-dong, Seoul 05505, Republic of Korea

**Keywords:** boronophenylalanine, boron neutron capture therapy, PK modeling, nonlinear mixed effect modeling

## Abstract

L-4-[(^10^B)]Boronophenylalanine (BPA) is an amino acid analogue with a boron-10 moiety. It is most widely used as a boron carrier in boron neutron capture therapy. In this study, a Bayesian predictive platform of blood boron concentration based on a BPA pharmacokinetic (PK) model was developed. This platform is user-friendly and can predict the individual boron PK and optimal time window for boron neutron capture therapy in a simple way. The present study aimed to establish a PK model of L-4-boronophenylalanine and develop a Bayesian predictive platform for blood boron PKs for user-friendly estimation of boron concentration during neutron irradiation of neutron capture therapy. Whole blood boron concentrations from seven previous reports were graphically extracted and analyzed using the nonlinear mixed-effects modeling (NONMEM) approach. Model robustness was assessed using nonparametric bootstrap and visual predictive check approaches. The visual predictive check indicated that the final PK model is able to adequately predict observed concentrations. The Shiny package was used to input real-time blood boron concentration data, and during the following irradiation session blood boron was estimated with an acceptably short calculation time for the determination of irradiation time. Finally, a user-friendly Bayesian estimation platform for BPA PKs was developed to optimize individualized therapy for patients undergoing BNCT.

## 1. Introduction

Radiation therapy aims to induce damage to cancer cells by delivering a large amount of ionizing radiation to the cancer tissue, which inevitably causes damage to normal tissues adjacent to the tumor in the radiation field. The treatment concept of boron neutron capture therapy (BNCT) differs fundamentally from conventional radiotherapy. Conventional photon and particle radiotherapy techniques aim to deliver radiation doses while conforming as much as possible to the target volume (i.e., gross tumor volume, clinical target volume, planning target volume, etc.) via external irradiation [1]. Currently, advanced radiotherapy techniques, such as intensity modulated radiation therapy (IMRT), volumetric modulated arc therapy (VMAT), proton therapy, heavy particle therapy, etc., have been introduced and are being constantly improved in order to more effectively irradiate the target volumes while mitigating radiation-induced damage in normal tissues [2,3,4].

Compared to current radiotherapies, which deliver radiation doses from external radiation via the ballistic approach, BNCT is considered a targeted radiation therapy, as it uses the ability of the ^10^B isotope to capture neutrons [5]. The irradiated high-energy neutrons gradually lose energy in tissues and becomes thermal neutrons. When one of these encounters a boron (^10^B) atom and causes a nuclear reaction, alpha particles and lithium (^7^Li) heavy ions are released. These alpha and (^7^Li) particles are high linear energy transfer (LET) particles producing large biological effects, with an average path length of 9 µm and 5 µm, respectively. Their short path range in tissues enables BNCT to target cancer cells with high boron concentration at the cellular level [6].

The radiation dose delivered by BNCT is affected by two factors, namely, neutron flux and boron concentration of each organ during the neutron irradiation; thus, precise monitoring of each component is crucial for the management of accurate radiation dose delivery. Compared to the ability to precisely detect and monitor the neutron flux, real-time estimation of boron concentration in each organ is hardly achievable, as access to the treatment room is restricted during neutron irradiation. In order to establish the radiation dose in each organ, we assumed the boron concentration of representative organs based on their organ-to-blood ratios as nominal values, considering the biodistribution of the boron drug as suggested in the previous literature based on an extensive review of preclinical and clinical data [7].

Previous studies have reported various methods for estimating blood boron levels after L-4-[(^10^B)]Boronophenylalanine infusion, the most commonly used boron carrier drug during irradiation. These methods include bi-exponential fitting [8] and two-compartment modeling-based estimation with bi-exponential fitting [9]. This study adopts NONMEM software version 7.4.3, which allows for analysis of a diverse range of pharmacokinetic (PK) and PD models accommodating a large number of parameters and is widely used in the pharmaceutical industry for PK modeling of various drugs. Furthermore, NONMEM incorporates uncertainty and inter-individual variabilities into its parameters, leading to more reliable PK and PD analyses with greater model flexibility. Additionally, NONMEM provides more detailed and accurate PK and PD estimates, aiding in efficient drug development.

In order for BNCT to deliver radiation doses as planned, precise PK modeling may be necessary in order for it to be recognized as a standardized and reproducible radiotherapy and to attract interest from radiation oncology researchers. The aim of this study was to develop a predictive platform for boron PK that can be rapidly and easily implemented during a clinical trial with high accuracy and precision through the use of NONMEM and R software version 4.2.3.

## 2. Materials and Methods

Our user-friendly predictive platform for boron PK was developed based on the following steps:Construction of a PK model for blood boron (^10^B)A user-friendly predictive interface consisting of a master R script, sub-R scripts for individual specific NONMEM dataset generation using the Shiny package, execution of NONMEM, and summation and graphical display of the NONMEM resultsSensitivity analyses to evaluate the predictive performance of the platform and identify the optimal PK sampling time for blood (^10^B)

### 2.1. PK Modeling of Blood (^10^B)

The target of the PK investigation was the temporal variation in boron ((^10^B)) concentration in blood post-intravenous administration of BPA. The dataset for analysis constituted 377 individual blood samples collected from an ensemble of 27 cancer patients who had undergone intravenous BPA infusion [8,10,11,12,13,14,15].

The modeling analysis of boron PK was conducted utilizing NONMEM version 7.4.3 (developed by ICON Development Solutions, Ellicott City, MD, USA) with gfortran compiler (version 4.4.0) in a Windows 10 environment. The NONMEM subroutine ADVAN13 and the first-order conditional estimation with interaction (FOCE-I) technique were adopted. The relationship between the population’s typical value and the individual’s PK parameter values was characterized using the equation Pi = PTV × exp(η), where PTV stands for the population’s typical value and η represented a random variable drawn from a normal distribution with zero mean and specific variance. During the modeling analysis, a conversion factor of 10/208.21 was applied to account for the molecular weight disparity between boron (^10^B)) and BPA.

### 2.2. Validation of Population PK Model

Various structural and error models were assessed, guided by a graphical assessment of optimum fit properties and statistical significance criteria. The validation of the population PK model was established by comparing the measured blood boron concentrations with the model’s predictions, thereby constituting its internal validation. Fundamental goodness-of-fit plots comparing observed and predicted concentrations were used to validate the model. Furthermore, the model was employed to generate population temporal concentration change predictions via 1000 Monte Carlo simulations, then the 95% prediction interval (the 2.5th through 97.5th percentiles) was compared with the actual concentrations in a visual predictive check (VPC). A likelihood ratio test was used to discriminate between hierarchic models at *p* < 0.05 based on the fact that the distribution of −2 log likelihood of the models approximately follows the chi-square distribution. In addition, the 95% confidence intervals for the estimated parameter values, derived assuming an a-normal distribution, were assessed to ensure that they did not contained any zero values.

### 2.3. Bayesian Predictive Model

The individual PK parameters for a new patient were determined using Maximum a Posteriori Probability (MAP) estimation, employing the ^10^B concentrations of the patient early after the beginning of BPA infusion and the NONMEM code of ^10^B PK model developed in this study. This was achieved by setting “maxeval = 0” in the Estimation step in NONMEM, allowing for prediction of changes in boron (^10^B) concentration across various dosage regimens. Markov Chain Monte Carlo (MCMC) Bayesian estimation was tested using $PRIOR, taking the PK model constructed in this study as the prior distribution model. The posterior probability distribution was obtained via Bayesian estimation of PK parameters using patient-specific ^10^B concentration data. A user-friendly interface was developed using R, the Shiny package, and NONMEM.

The platform consists of a data generation tool that creates a csv file in NONMEM data format, a Bayesian prediction tool that uses NONMEM input data to produce patient-specific boron (^10^B) PK predictions, and a recommendation tool that suggests neutron irradiation time based on predicted blood boron concentration. The interface is designed for ease of use, helping to streamline the implementation of the PK model and Bayesian estimation. The platform primarily utilizes NONMEM for Bayesian PK estimation, R for data generation in NONMEM format using the Shiny package and NONMEM execution, and summarization and graphical display of the NONMEM outputs. The end results, compiled into a pdf file, enable the identification of a patient’s PK profile with numerical information and assist in determining the neutron irradiation time.

### 2.4. Optimal Sampling Schedule Determination for Blood Boron Concentration Prediction

To optimize blood sampling time points for predicting ^10^B concentration, we simulated ^10^B concentration over time following the intravenous infusion of BPA at 500 mg/kg over a 180 min with 1000 replicates. Using the PK model constructed in this study, we simulated ^10^B concentrations at specific blood boron concentration points, as shown in Table 1, and compared the concentrations during irradiation with the simulated concentrations. The predictability of Bayesian approach was then evaluated in terms of accuracy and precision.

The Maximum a Posteriori Probability (MAP) and Markov Chain Monte Carlo (MCMC) Bayesian methods were utilized to generate time-dependent concentration values for each individual subject. Simultaneously, predicted mean values for blood boron concentration were produced within a 240–300-min window, which was the selected time point for neutron irradiation. These values were compared to the initially simulated mean blood boron concentration. The entire process was repeated 1000 times for each blood sampling schedule listed in Table 1, which enabled us to assess the accuracy and precision of the Bayesian prediction method for each specific blood sampling schedule. To measure prediction accuracy, we compared the true concentration value in the simulation, represented by IPRED and IPREDCAVR, to the true value predicted by the model using Bayesian prediction, represented by RIPRED and RIPREDCAVR, with the following equations:PE=IPRED−RIPRED,
%PE=100 × ((IPRED−RIPRED)/RIPRED),
PECAVR=IPREDCAVR−RIPREDCAVR,
%PECAVR=100 × ((IPREDCAVR−RIPREDCAVR)/RIPREDCAVR).


In the above equations, PE refers to the prediction error, IPRED represents the individual model predicted value, RIPRED represents the simulated true value or the model prediction value that includes the blood boron concentration value at 300 min, PECAVR denotes the prediction error for the interval mean concentration value, and RIPREDCAVR signifies the simulated value or the model prediction value that includes the interval mean value. The goal was to validate a blood sampling schedule that could provide an accuracy within 5% with respect to the blood boron concentration value inclusive of the 300 min time point.

## 3. Results

### 3.1. PK Modeling

The parameter estimates were systematically evaluated and are detailed in Table 2. The boron concentration in the blood over time was effectively captured by a two-compartment linear model. The residual model was aptly fit using a mixed additive and proportional model. The observed concentration (Cobs) was well described by the equation below:Cobs=Cpred+ϵ1+ϵ2×Cpred
Cobs, observed concentration; Cpred, concentration predicted by the model; ϵ1 and ϵ2, random variables with a normal distribution with mean zero).

Our PK model showed strong predictive power without significant bias in its evaluation. Employing a nonlinear mixed effects model, the data were effectively processed through fixed and random effects. Residuals representing the discrepancy between observed and model-predicted values consisted of additive and proportional portions. The model’s predicted values for population blood concentration and individual predicted values are based on the fixed effect alone and in conjunction with individual differences in the random effect, respectively.
pharmaceuticals-17-00301-t002_Table 2Table 2Results of blood boron (B10) pharmacokinetic modeling during intravenous administration of B10-BPA.ParameterEstimates95% Confidence IntervalK12, 1/min0.0230.018–0.028IIVK12 (CV %)0.071 (27.2)0.005–0.137K21, 1/min0.0120.009–0.014IIVK21 (CV %)0.037 (19.4)0.012–0.061K10, 1/min0.0060.005–0.007IIVK10 (CV %)0.050 (22.6)−0.002–0.102V1, L0.2520.216–0.288IIVV1 (CV %)0.061 (25.0)0.029–0.092ϵ1 (additive)*, μg/g0.5400.054–1.026ϵ2 (proportional)*0.0010.000–0.003

The model’s suitability was further confirmed by the balanced and unbiased distribution of data points surrounding the baseline in the basic goodness-of-fit plot (Figure 1A–D) and the evenly distributed residuals in the Conditional Weighted Residual plot. Comparing the predicted and observed concentration values over time (Figure 1E) reveals the model’s accurate prediction of individual observed values across the entire time range (Appendix A). In addition, the visual predictive check (Figure 2) demonstrated high predictive accuracy, with most observed boron (^10^B) concentration values falling within the model’s 95% prediction range. Furthermore, the 95% confidence interval of all fixed and random effect parameters constructed based on the point estimates and their respective standard errors did not include zero, underlining the model’s statistical validity.

### 3.2. Optimal Sampling Schedule Determination

Both the Maximum a Posteriori (MAP) Bayesian method and the Markov Chain Monte Carlo (MCMC) Bayesian method showed high accuracy in predicting population averages for boron concentrations over time and during neutron exposure times of 240–300 min. However, prediction precision showed a decrease over time, with the 95% prediction interval of the prediction error expanding as time advanced (Figure 3A), and enhancing the frequency of boron concentration measurements improved the prediction accuracy, especially when data near the neutron exposure start time of 240 min were included (Figure 3B).

Upon comparing different blood sampling schedules (Table 1), we found that using nine blood boron concentration data points—three samples taken hourly post-administration and six additional samples at intervals of 10 min up to 240 min—could predict the average blood boron ((^10^B)) concentration during neutron exposure within a ±5% error range of the 95% prediction interval when using the MAP Bayesian prediction method (Figure 3C).

### 3.3. User-Friendly Prediction Platform for Boron Concentration Estimation

Due to the time-intensive nature of calculating the radiation dose per average blood boron concentration prior to irradiation, recalculating the radiation dose and irradiation time based on the real-time boron concentration is not practically feasible. This complexity is further compounded by the gradual decrease in blood boron concentration over time, which necessitates accurate determination of the neutron irradiation time in order to achieve a specified dose considering the average blood boron concentration after the irradiation starts. To overcome this, we propose a strategy wherein the necessary neutron irradiation time for each blood boron concentration range is pre-checked during the treatment planning phase. This phase includes determining the treatment duration, direction, and posture. When requiring an irradiation time within the predicted range (95% confidence interval of average boron concentration during irradiation, which was assumed to be 10–30 ppm based on the population PK model), the corresponding irradiation time can be more realistically determined, providing a practical solution for BNCT. On the day of BNCT treatment, and after administering BPA, boron concentrations are input at each collection point via blood sampling to predict the average blood boron concentration at the time of neutron irradiation (a screen capture of the data input interface is presented in Appendix A). By applying regression analysis with the pre-checked irradiation times, our approach offers swift and precise determination of neutron irradiation times to assist researchers, as demonstrated in (Figure 4).

## 4. Discussion

BNCT is a precise radiotherapy that utilizes the ability of boron-10 to capture thermalized neutrons, leading to a subsequent fission reaction that emits alpha and Li-7 particles, which deliver high-LET (linear energy transfer) radiation to tissues at the cellular level. In BNCT, the precision of treatment planning and delivery is dependent upon two key factors: the amount of neutrons, and the boron concentration in each tissue. These parameters must be accurately determined or estimated to ensure effective treatment.

The accuracy of radiotherapy delivery is essential to achieving the desired balance between tumor control (TCP) and minimizing normal tissue complications (NTCP) as demonstrated by dose response curves. In an effort to optimize radiotherapy outcomes and advance treatment techniques, it is imperative to consider the steepness of the TCP or NTCP curve as a key factor in determining the required accuracy of dose delivery. Any deviation from the planned dose, even in small amounts, can result in a decrease in TCP or an increase in NTCP, with the steepest curves being observed for normal tissue effects with a γ50 value of up to 6–7% per 1% change in dose; this highlights the importance of ensuring accurate dose delivery in clinical practice [16].

According to the International Commission on Radiation Units and Measurements (ICRU), Report 24, the recommended accuracy for radiotherapy delivery should be 5% or less. Other studies, such as those conducted by Mijnheer et al. and Brahme et al., have reported required accuracy levels by considering normal tissue complications and the impact on TCP, with values ranging from 7% to 3% relative standard deviation, respectively. These findings underscore the need for ongoing efforts to improve accuracy in radiotherapy delivery and maintain the balance between maximizing tumor control and minimizing tissue complications.

In BNCT, multiple quality assurance items are performed to ensure an accurate radiation dose, including verification of dose calculation algorithms, validation of treatment planning systems, and examination of the physical characteristics of the neutron beam of the therapeutic neutron device. Even with the implementation of various quality assurance measures of the physical properties of neutrons and devices, the challenge of accurately estimating and monitoring changes in blood boron concentration caused by boron pharmaceuticals in vivo remains as an outstanding issue of uncertainty. Previous attempts to address this challenge have involved using linear estimation or a combination of two-compartment and biphasic estimation with spreadsheet tools, resulting in high accuracy.

In a clinical study conducted at Brookhaven National Laboratory in the late 1990s [17], the average blood concentration was calculated by linear extrapolation of blood concentrations just before, during, and after neutron irradiation in the nuclear reactor. From 1996 to 1999, the Harvard Institute of Technology and MIT conducted a clinical study jointly [9] revealing that the blood concentration could be modeled by reference to the fact that the existing literature on blood boron concentration showed a bi-exponential washout. Prediction and post-testing were successfully performed on this basis. In Finland, a two-compartmental model and bi-exponential model were constructed using the patient blood concentration of the previous study conducted at the Brookhaven National Laboratory and predicted on this basis [8]. A sponsor-led clinical trial by Japan’s Stellar Pharma and Sumitomo Heavy Industries resulted in product approval as a BNCT for recurrent head and neck cancer. Another the sponsor-led clinical trial was conducted in glioblastoma, with the time corresponding to a specific dose based on the blood concentration immediately before neutron irradiation. However, a separate PK model for the prediction of blood boron concentration during neutron irradiation was not disclosed.

In this study, we propose a more advanced and reproducible simulation method using NONMEM, a widely accepted tool in population PK and PD modeling, to estimate blood boron concentration during neutron irradiation. To do this, we extract data through graphical analysis using individual blood boron concentration data from clinical trials and establish a PK model as an a priori model using these raw data. Based on this model, we perform Bayesian prediction using a nonlinear mixed-effects model to predict the blood boron concentration during neutron irradiation. Through this approach, we select the number of blood samplings and the timing of samplings within 5% of the predicted blood boron concentration with 95% confidence before neutron irradiation. This approach allows for more precise and individualized prediction of boron concentration.

The use of blood boron concentration as a means to estimate tissue concentration has certain limitations. The measurement of ^10^B concentration in venous blood through sampling may not be a sufficiently reliable approach for estimating the ^10^B concentration in multiple tissues and organs, considering that the precise ratios between blood and tissues have not been fully characterized. It is not possible to accurately assess the ratio of each layer or tissue within the radiation field; in addition, it is not feasible to obtain biopsies of every individual cell or tissue in order to directly measure its boron concentration, which can vary among individuals (inter-individual variation). Thus, in order to facilitate practical application, a certain nominal value has been proposed and utilized for this purpose in previous clinical trials.

In the early phase of clinical studies of BNCT, the central nervous system (CNS), particularly the brain, was a primary concern with regard to the dose-limiting toxicity of therapy for the treatment of gliomas. The concentration for normal nervous tissues, including the brain, was estimated based on the intravascular boron concentration in the blood. The results of subsequent biodistribution studies indicated that the intravascular boron concentration in the blood was slightly higher than that in the brain tissue [18,19] due to the fact that the blood–brain barrier (BBB) restricts the entry of boron into the brain parenchyma. This makes the blood a good proxy for the intravascular boron concentration in the brain. Additionally, damage to the vascular endothelium in the brain could result in brain edema; therefore, the higher boron concentration in the blood was used to calculate the standard for the brain in [7]. Other normal tissues, such as mucosa and skin, have been estimated based on the intravascular boron concentration in the blood as well.

Regarding the boron concentration in the tumor, Elowitz et al. reported the results of a biodistribution study using BPA which demonstrated a range of tumoral uptake values [18]. Coderre et al. subsequently reported that the ^10^B concentration in the gross tumor sample varied from less than 10 ug ^10^B/g to more than 60 ug ^10^B/g when normalized to a BPA dose of 250 mg/kg [19]. To account for the variability in tumor cellularity, a nominal value of 3.5, considered a conservative estimate, was proposed, and has been widely accepted in clinical trials. This approach was adopted to estimate the tumoral dose in a practical manner by assuming that tumoral cells have at least 3.5 times higher uptake of BPA compared to the intravascular boron concentration in the blood.

BNCT-SPECT [20,21,22] and radioisotope imaging modalities such as 4-^10^B-Borono-2-^18^F-fluoro-L-phenylalanine (^18^F-FBPA) [23,24,25,26] offer a promising means of determining individual tumor-to-blood ratios (TBRs) for BNCT by providing detailed information on the distribution of boron within a patient’s tumor tissue. It should be noted that this imaging-based approach provides an average TBR value across the entire gross tumor volume, which may be influenced by variables such as tumor cellularity. Nevertheless, ongoing research is exploring the potential of deriving individual TBRs of BPA in normal and tumor tissues using ^18^F-FBPA, which could further enhance the precision and efficacy of BNCT for patients with various malignancies.

As an alternative, the use of a conservative nominal value, such as the 3.5 TBR proposed by Coderre et al. and which has been widely accepted in clinical trials for BNCT, is based on the assumption that tumoral cells have at least 3.5 times higher uptake of BPA compared to the intravascular boron concentration in the blood, which is being rapidly measured in real-time [19]. While this approach may not always be appropriate for individual patients with various heterogeneities, it provides a practical and feasible solution for determining the tumoral dose in standardized clinical practice.

The reason for including the sampling time points in the current study for the assessment of boron concentration during neutron irradiation is that it may not be feasible for personnel to access the high-radiation field in order to obtain a blood sample during irradiation. Thus, it is necessary to establish a platform that allows for quick, reproducible, and reliable estimation of blood boron concentration by investigators during irradiation. While this may vary depending on the location of the tissue or organs within the radiation field, it is estimated to take up to an hour of irradiation time. Therefore, our focus has been to estimate the mean blood boron concentration with as much accuracy as possible, with a precision of less than ±5% (95% confidence interval) during an hour of irradiation. The raw data used in this study were derived from graphical analysis of the previous literature, which may have resulted in a substantial residual, leading to the need for additional sample time points. However, this limitation can be addressed through remodeling of the PK based on the data collected during the ongoing clinical trial, reducing the number of sample time points required in future iterations.

## 5. Conclusions

In this project, a PK model based on nonlinear mixed-effects modeling was constructed and an optimal method for predicting the PK of intravenous blood boron (^10^B) in the case of intravenous administration of BPA was established by the Bayesian method. Based on this, it was confirmed by analyzing the distribution of boron concentrations in blood and tumors over time that the sequential irradiation method (a method of irradiating neutrons after the end of drug administration) is more appropriate than the simultaneous irradiation treatment previously attempted in Japan.

As a representative sequential irradiation method, it is possible to predict the blood boron concentration with high accuracy and precision using this prediction platform by performing sensitivity analysis through simulation using the method of injecting 500 mg/kg for 3 h and irradiating neutrons one hour later. The time of blood collection and amount of blood that can be drawn were identified. After the start of administration, blood was drawn three times at one-hour intervals (60 min, 120 min, 180 min) and six additional times up to 240 min thereafter at intervals of 10 min. It was confirmed that this approach can be used to predict blood ^10^B concentrations with high reliability and accuracy.

In addition, we developed a user-friendly prediction platform that can be applied to deliver precise BNCT doses in actual clinical sites. The prediction platform was able to produce stable prediction results in many experimental runs, justifying its application in actual clinical fields such as clinical trials. The predicted platform constructed in this study can help to evaluate the therapeutic effect of BNCT with improved accuracy and reliability in actual clinical trials, and could contribute significantly to the efficient development of novel therapies.

## Figures and Tables

**Figure 1 pharmaceuticals-17-00301-f001:**
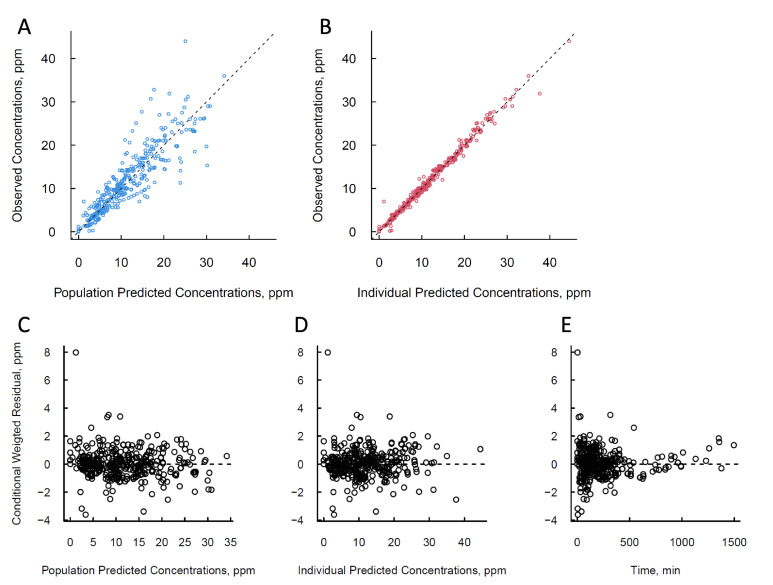
Basic goodness of fit plot of the population pharmacokinetic model after intravenous infusion of ^10^B-BPA. The fits for the observed value versus the population predicted concentrations and the observed value versus the individual predicted concentrations were judged to be good, and the trend line was close to the line of unity. (**A**) Observed concentration versus population predicted concentration. (**B**) Observed concentration versus individual predicted concentration. (**C**) Conditional weighted residuals versus population predicted concentration. (**D**) Conditional weighted residuals versus individual predicted concentration. (**E**) Conditional weighted residuals versus time after the intravenous infusion.

**Figure 2 pharmaceuticals-17-00301-f002:**
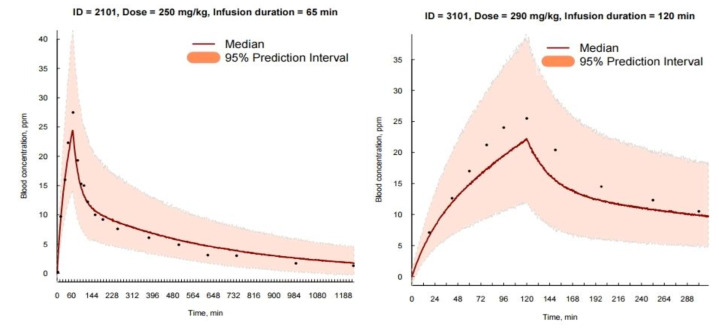
Case example of individual visual predictive check (VPC). Observed concentration over time versus median model prediction and 95% prediction interval. The VPC demonstrated good agreement between the observed values in patients and the predicted values.

**Figure 3 pharmaceuticals-17-00301-f003:**
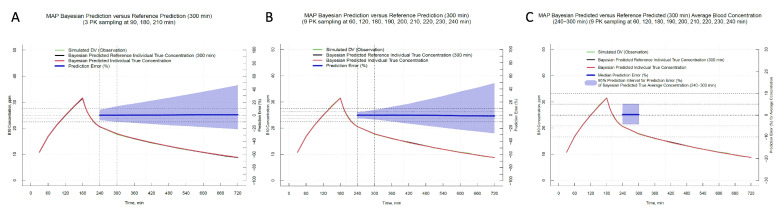
(**A**,**B**) display the prediction error over time (240 to 720 min post-BPA administration), revealing a gradual increase in the discrepancy between the predicted and reference values. (**C**) highlights the average prediction error between 240 and 300 min post-injection, with the 95% confidence interval confined within 5%. The green line represents the simulated Dependent Variable (DV), which is the mean of the bootstrap values from the Bayesian Predicted Reference Individual True Concentration (IPRED, depicted by the black line) with the addition of residuals to reflect observed data. The Bayesian Predicted Individual True Concentration, represented by the red line, is calculated from the mean bootstrap values of estimated concentrations based on the simulated DV at the specified timepoints. In (**A**), these timepoints are 90, 180, and 210 min post-infusion, while in (**B**,**C**) they are 60, 120, 180, 190, 200, 210, 220, 230, and 240 min post-infusion.

**Figure 4 pharmaceuticals-17-00301-f004:**
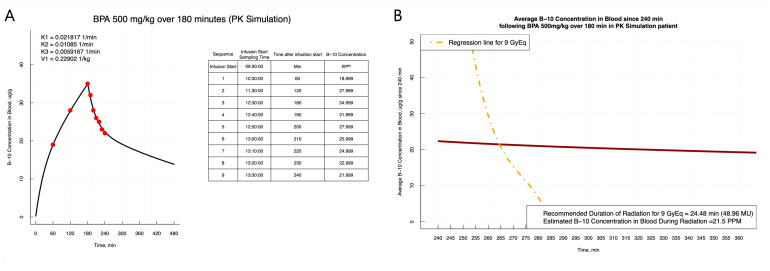
Platform for boron concentration estimation based on Bayesian prediction of individual pharmacokinetic parameters and subsequent determination of neutron irradiation duration. (**A**) The measured individual ^10^B concentration, indicated by the red dot, is used for subsequent estimation of pharmacokinetic parameters and ^10^B concentration. The table on the right shows the detailed blood sample information. (**B**) Average ^10^B concentration after irradiation, shown by the bold red line, and required irradiation time, shown by dash-single dotted line. The box in the lower right shows the required neutron irradiation duration and the expected boron ^10^B concentration during neutron irradiation.

**Table 1 pharmaceuticals-17-00301-t001:** Blood sampling schedule used to determine optimal collection frequency and timing for blood boron (^10^B) concentration prediction.

No.	Number of Blood Samplings and Timepoints per Schedule
I	3 PK samplings at 90, 180, 210 min
II	4 PK samplings at 90, 180, 210, 240 min
III	4 PK samplings at 90, 180, 200, 220 min
IV	5 PK samplings at 90, 180, 200, 220, 240 min
V	4 PK samplings at 90, 180, 195, 210 min
VI	5 PK samplings at 90, 180, 195, 210, 240 min
VII	6 PK samplings at 90, 180, 190, 200, 210, 220 min
VIII	8 PK samplings at 90, 180, 190, 200, 210, 220, 230, 240 min
IX	4 PK samplings at 60, 120, 180, 210 min
X	5 PK samplings at 60, 120, 180, 210, 240 min
XI	5 PK samplings at 60, 120, 180, 200, 220 min
XII	6 PK samplings at 60, 120, 180, 200, 220, 240 min
XIII	5 PK samplings at 60, 120, 180, 195, 210 min
XIV	7 PK samplings at 60, 120, 180, 195, 210, 225, 240 min
XV	7 PK samplings at 60, 120, 180, 190, 200, 210, 220 min
XVI	9 PK samplings at 60, 120, 180, 190, 200, 210, 220, 230, 240 min

## Data Availability

Research data are not available at this time as the data are the intellectual property of the sponsor.

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
