# Peer review of "PK Modeling of L-4-Boronophenylalanine and Development of Bayesian Predictive Platform for L-4-Boronophenylalanine PKs for Boron Neutron Capture Therapy"

_pharmaceuticals, 2024, doi:10.3390/ph17030301_

Round 1

Reviewer 1 Report

Comments and Suggestions for Authors

I think the paper titled "PK Modeling of L-4-boronophenylalanine and Development of Bayesian Predictive Platform for L-4-boronophenylalanine PKs for Boron Neutron Capture Therapy" is very interesting and novel. My comments are below.

#1  P.10 L 338-342

In conclusion, blood sampling at 9 time points is required.

If so many blood samples are to be taken, I don't think there is a need for predictions using a model.

Please indicate that predictions by the PK model are required.

#2  P.2 L65

Although it says "The user-friendly Bayesian estimation platform", I think the form consisting of NONMEM and R cannot be called user-friendly. 

Also, is this platform bug-free?

#3 P.5 L153

In addition to drug characteristics, blood sampling timing and infusion time contribute to determining the compartment model, so the blood concentration time curve is important information to evaluate whether the model is appropriate. Please show the actual blood concentration-time curve.

Then, please explain why you did not consider other compartment models, such as a 1-compartment or 3-compartment model. Or please show the AIC value to justify your model selection.

#4 P.2 L 72

When applying the PK model of this study to real patients, patient background is important.

Therefore, please indicate the patient background of the dataset.

#5 P.8 Figure 4.

The explanation of Figure 4 is difficult to understand. Please explain clearly.

Author Response

Dear Reviewer #1

Thank you for your feedback on our manuscript. We value your insights and have addressed each of your comments as follows:

**Response to Comment 1:**

You raised an important point about the need for PK model predictions using nine blood samples. Here's why we think the model is still necessary:

-It accurately predicts boron levels at times when we can't take samples, like during irradiation, which enables us to estimate “average blood boron concentration” during the irradiation. If the blood boron during the irradiation is 20ppm, it requires, let’s say, 25 minute is required for the planned dose delivery. But still, we need to calculate “the average blood boron concentration” after initiation of neutron irradiation and the duration of neutron irradiation needs to be decided. If the average blood boron during the irradiation is expected to be 18.3 ppm, for example, then we may need to recalculate the irradiation time needed for the decision of irradiation time in a real time manner. This platform helps us by estimating “average blood boron concentration” from a certain time point (initiation of irradiation) and with timely suggestion to reach the predefined targeted level of radiation exposure.

- The model sets the stage for future studies where we might reduce the number of needed samples. The pharmacokinetic model was constructed based on the sample information form literatures and it may bear quite a large size of residual, which means it requires somewhat larger sample numbers for accurate estimation for now. This will be reduced per assessment of internal residual of the samples acquired from the subjects in the clinical trial and will be reduced soon after the analysis.

**Response to Comment 2:**

It's designed using R Shiny tool and it is not a code-free work, but the whole process is a quite seamless process since the user may just enter the relevant values in the each intended cells. R shiny based Pharmacokinetic estimation with NONMEM and R and simplifies their work.

As for being bug-free, we've tested it extensively, internally validated qualitatively using the raw data which has been used for model development. We have internal Standard Operating Procedure (SOP) for installation qualification and operational qualification to prevent any bias, accidental errors or bugs during process. We understand that complete certainty is hard to achieve in software. We'll keep improving it over time.

**Response to Comment 3:**

Thanks for asking about the blood concentration-time curve and model choice.

Actual blood concentration-time curve is displayed as a component of figure 2.

 We chose the two-compartment based on graphical and statistical critieria. It was also shown to best fit the pharmacokinetics of the BPA in the previous literature.  

To be clear, we added the following sentences in the method section.

“Various structural and error models were assessed, guided by a graphical assessment of optimum fit properties and statistical significance criteria.”

“A likelihood ratio test was used to discriminate between hierarchic models at p<0.05, based on the fact that the distribution of -2 log likelihood of the models follows approximately chi square distribution.”

And, we did not conduct non-parametric bootrap for this PK modeling, and we removed the description about the bootstrap from this manuscript.

**Response to Comment 4:**

In our model, the patient demographics were drawn from a dataset that consisted of 27 cancer patients from various studies. It's worth noting that many of these studies did not provide comprehensive demographic data as in the below table.

Reference

Demographic Information Available

Analytical Method

Ryynänen PM, Kortesniemi M, Coderre JA, Diaz AZ, Hiismäki P, Savolainen SE. Models for estimation of the 10B concentration after BPA-fructose complex infusion in patients during epithermal neutron irradiation in BNCT. Int J Radiat Oncol Biol Phys. 2000;48(4):1145-1154. doi:10.1016/S0360-3016(00)00766-5

Demographics of the subjectnot described

direct current plasma-mass spectrometry or prompt gamma neutron activation analysis.

Kiger WS, Palmer MR, Riley KJ, Zamenhof RG, Busse PM. Pharamacokinetic modeling for boronophenylalanine-fructose mediated neutron capture therapy: 10B concentration predictions and dosimetric consequences. J Neurooncol. 2003;62(1-2):171-186. doi:10.1023/A:1023297107525

Age, gender and weight disclosed in the text, other details undisclosed.

Prompt Gamma Neutron Activation Analysis (PGNAA) and/or Inductively Coupled Plasma Atomic Emission Spectroscopy (ICP-AES)

Kiger WS, Palmer MR, Riley KJ, Zamenhof RG, Busse PM. A pharmacokinetic model for the concentration of 10B in blood after boronophenylalanine-fructose administration in humans. Radiat Res. 2001;155(4):611-618. doi:10.1667/0033-7587(2001)155[0611:apmftc]2.0.co;2

Gender disclosed in the text, other details undisclosed.

prompt gamma neutron activation analysis and/or inductively coupled plasma atomic emission spectroscopy (ICP-AES)

Zhong Y, Zewen S, Yongmao Z, et al. Boron neutron capture therapy for malignant melanoma: first clinical case report in China. Chinese J Cancer Res. 2016;28(6):634-640. doi:10.21147/j.issn.1000-9604.2016.06.10

Gender disclosed in the text, other details undisclosed.

inductively coupled plasma atomic emission spectrometry (ICP-AES).

Wittig A, Collette L, Appelman K, et al. EORTC trial 11001: Distribution of two 10B-compounds in patients with squamous cell carcinoma of head and neck, a translational research/phase 1 trial. J Cell Mol Med. 2009;13(8 B):1653-1665. doi:10.1111/j.1582-4934.2009.00856.x

Age, gender, weight and height available for three subjects, other details undisclosed.

10B-concentrations were measured with prompt gamma ray spectroscopy (PGRS).

Fukuda H, Hiratsuka J, Honda C, et al. Boron neutron capture therapy of malignant melanoma using 10B-paraboronophenylalanine with special reference to evaluation of radiation dose and damage to the normal skin. Radiat Res. 1994;138(3):435-442. doi:10.2307/3578693

Age and gender of the Subjects are available, other details undisclosed.

10B concentration was measured by a chemical method, inductively coupled plasma atomic emission spectrometry (ICP-AES) or prompt y-ray analysis

Koivunoro H, Hippeläinen E, Auterinen I, et al. Biokinetic analysis of tissue boron (10B) concentrations of glioma patients treated with BNCT in Finland. Appl Radiat Isot. 2015;106:189-194. doi:10.1016/j.apradiso.2015.08.014

Demographics of the subjectnot described

inductively coupled plasma-atomic emission spectroscopy (ICP- AES)

However, when evaluating the model's ability to handle residual tolerance and the required sample size for achieving accuracy and generalizability, the inclusion of blood sample data from a diverse population with different interindividual variability serves as a safeguard against potential bias in the estimation platform.

**Response to Comment 5:**

Regarding the clarity of the Figure lends of the Fig. 4, Please find the below updated figure legend.

Figure 4. Screenshot of the Platform for average boron concentration during neutron irradiation estimated from individual blood samples and subsequent determination of neutron irradiation duration. (A) Measured 10B concentration (red dots) with estimated 10B concentration trend (in black line) with sampling time and the results of each sample in numbers in table. (B) This panel illustrates the average 10B boron concentration during neutron irradiation (depicted as a bold red line), which is critical for dose calculation in BNCT. The graph demonstrates a relative decrease in the steepness of the concentration slope compared to the estimated individual 10B concentration trend shown in Fig. 4A, reflecting the average concentration after the initiation of irradiation. The dashed-single dotted line indicates the required neutron irradiation duration, which is determined through a regression analysis based on various assumed blood boron concentrations (ranging from 10 to 30 ppm). This analysis facilitates the estimation of the necessary irradiation time to achieve a predefined therapeutic dose (e.g., a brain maximum dose of 11 GyEq or 9 GyEq), tailored to the patient's specific boron concentration on the day of treatment. For instance, if the patient exhibits an average blood boron concentration of 22 ppm, the platform predicts a slightly extended irradiation duration compared to the base assumption of 20 ppm. This approach allows for rapid, patient-specific adjustment of irradiation times in real-time, ensuring precise and effective BNCT dosage administration. The lower right box in the figure displays the calculated neutron irradiation duration and the expected average 10B concentration during this period.

Also, please find the revised manuscript in response to the reviewer's comments.

We hope these responses address your concerns and help improve our manuscript. Your feedback is invaluable to us.

Best regards,

Woo

Reviewer 2 Report

Comments and Suggestions for Authors

The manuscript develops a pharmacokinetic (PK) model of L-4-boronophenylalanine and develop a Bayesian predictive platform for blood boron PKs for user-friendly estimation of boron concentration during neutron irradiation of neutron capture therapy. Data were collected retrospectively from seven previous reports. Moreover, model robustness was assessed using nonparametric bootstrap and visual predictive check approaches. In general, the structure and concise wording of the manuscript are adequate, which greatly facilitates its understanding by potential readers. On the other hand, tables and figures in the manuscript are adequate.

Comment 1: The dose is not specified in the manuscript and the data were collected from seven previous reports. What dose was administered in each report? Was it the same in all reports? I would appreciate it if you could include the dosing schedule in each report. Additionally, the dose depends on the weight. Are the demographic characteristics of the population of each study similar?

Comment 2: Is the analytical technique the same in all the reports?

Comment 3: I would appreciate it if you could include the CV% for additive and proportional errors and for K12 in Table 2.

Author Response

Dear Reviewer #2

Thank you for your feedback on our manuscript. We value your insights and have addressed each of your comments as follows:

**Response to Comment 1:**

: Thank you for your insightful query regarding the dosing schedules and demographic characteristics of the populations in the seven previous reports referenced in our study.

Each of these studies, as listed below, had its unique dosing schedule(with a range of 100-450 mg/kg and infusion durations varying from 1-5 hours), reflecting a diverse approach to boron neutron capture therapy (BNCT) across different institutions and patient demographics.

Reference articles

Study 1: Ryynänen, P. M., Kortesniemi, M., Coderre, J. A., Diaz, A. Z., Hiismäki, P., & Savolainen, S. E. (2000). Models for estimation of the 10B concentration after BPA-fructose complex infusion in patients during epithermal neutron irradiation in BNCT. International Journal of Radiation Oncology Biology Physics, 48(4), 1145–1154. https://doi.org/10.1016/S0360-3016(00)00766-5a

Study 2: Kiger WS, Palmer MR, Riley KJ, Zamenhof RG, Busse PM. Pharamacokinetic modeling for boronophenylalanine-fructose mediated neutron capture therapy: 10B concentration predictions and dosimetric consequences. J Neurooncol. 2003;62(1-2):171-186. doi:10.1023/A:1023297107525

Study 3: Kiger WS, Palmer MR, Riley KJ, Zamenhof RG, Busse PM. A pharmacokinetic model for the concentration of 10B in blood after boronophenylalanine-fructose administration in humans. Radiat Res. 2001;155(4):611-618. doi:10.1667/0033-7587(2001)155[0611:apmftc]2.0.co;2

Study 4: Zhong Y, Zewen S, Yongmao Z, et al. Boron neutron capture therapy for malignant melanoma: first clinical case report in China. Chinese J Cancer Res. 2016;28(6):634-640. doi:10.21147/j.issn.1000-9604.2016.06.10

Study 5: Wittig A, Collette L, Appelman K, et al. EORTC trial 11001: Distribution of two 10B-compounds in patients with squamous cell carcinoma of head and neck, a translational research/phase 1 trial. J Cell Mol Med. 2009;13(8 B):1653-1665. doi:10.1111/j.1582-4934.2009.00856.x

Study 6: Fukuda H, Hiratsuka J, Honda C, et al. Boron neutron capture therapy of malignant melanoma using 10B-paraboronophenylalanine with special reference to evaluation of radiation dose and damage to the normal skin. Radiat Res. 1994;138(3):435-442. doi:10.2307/3578693

Study 7: Koivunoro H, Hippeläinen E, Auterinen I, et al. Biokinetic analysis of tissue boron (10B) concentrations of glioma patients treated with BNCT in Finland. Appl Radiat Isot. 2015;106:189-194. doi:10.1016/j.apradiso.2015.08.014

Regimen in representative cases in the reference articles have not been included in the articles which hinders further discussion regarding the baseline differences in between the demographic information.

Moreover, these reports were conducted by different institutions worldwide, and as such, the study populations may not be homogenous. Detailed demographic characteristics for each study were not consistently provided in the original reports, which limits our ability to make direct comparisons or generalizations about the patient populations.

In our research, we have developed a comprehensive pharmacokinetic (PK) model using NONMEM software. This model is specifically designed to address the variability found in different institutions and the diversity in patient demographics, which are common challenges in Boron Neutron Capture Therapy (BNCT) trials. Our approach effectively integrates a wide spectrum of dosing regimens and patient characteristics into the residual analysis, thereby enhancing the relevance and applicability of our findings across a variety of clinical contexts.

Considering the significant variation in dosing and infusion times reported and potential demographic differences in the referenced studies, we made a deliberate decision to exclude detailed dosing schedules, regimen, and demographics of the raw data from our manuscript. We are of the opinion that such detailed inclusion could potentially detract from the primary focus of our study. The core objective is to introduce and elucidate the development of a prediction platform for BNCT trials that is not only user-friendly and accurate but also universally applicable.

We believe this approach aligns with the standards of the review process and are receptive to any additional guidance or queries the reviewers may have on this matter.

**Response to Comment 2:**

: We appreciate your question regarding the uniformity of analytical techniques across the seven reports. It is important to acknowledge that the analytical methods employed to measure boron concentrations in these studies did indeed vary. Within the literatures that have referenced for the modeling, some institutions utilized one or more techniques among ICP-AES (Inductively Coupled Plasma Atomic Emission Spectroscopy), ICP-OES (Inductively Coupled Plasma Optical Emission Spectroscopy), or ICP-MS (Inductively Coupled Plasma Mass Spectrometry) as below. The table also details the inclusion of demographic information.

Reference

Demographic Information Available

Analytical Method

Ryynänen PM, Kortesniemi M, Coderre JA, Diaz AZ, Hiismäki P, Savolainen SE. Models for estimation of the 10B concentration after BPA-fructose complex infusion in patients during epithermal neutron irradiation in BNCT. Int J Radiat Oncol Biol Phys. 2000;48(4):1145-1154. doi:10.1016/S0360-3016(00)00766-5

Demographics of the subjectnot described

direct current plasma-mass spectrometry or prompt gamma neutron activation analysis.

Kiger WS, Palmer MR, Riley KJ, Zamenhof RG, Busse PM. Pharamacokinetic modeling for boronophenylalanine-fructose mediated neutron capture therapy: 10B concentration predictions and dosimetric consequences. J Neurooncol. 2003;62(1-2):171-186. doi:10.1023/A:1023297107525

Age, gender and weight disclosed in the text, other details undisclosed.

Prompt Gamma Neutron Activation Analysis (PGNAA) and/or Inductively Coupled Plasma Atomic Emission Spectroscopy (ICP-AES)

Kiger WS, Palmer MR, Riley KJ, Zamenhof RG, Busse PM. A pharmacokinetic model for the concentration of 10B in blood after boronophenylalanine-fructose administration in humans. Radiat Res. 2001;155(4):611-618. doi:10.1667/0033-7587(2001)155[0611:apmftc]2.0.co;2

Gender disclosed in the text, other details undisclosed.

prompt gamma neutron activation analysis and/or inductively coupled plasma atomic emission spectroscopy (ICP-AES)

Zhong Y, Zewen S, Yongmao Z, et al. Boron neutron capture therapy for malignant melanoma: first clinical case report in China. Chinese J Cancer Res. 2016;28(6):634-640. doi:10.21147/j.issn.1000-9604.2016.06.10

Gender disclosed in the text, other details undisclosed.

inductively coupled plasma atomic emission spectrometry (ICP-AES).

Wittig A, Collette L, Appelman K, et al. EORTC trial 11001: Distribution of two 10B-compounds in patients with squamous cell carcinoma of head and neck, a translational research/phase 1 trial. J Cell Mol Med. 2009;13(8 B):1653-1665. doi:10.1111/j.1582-4934.2009.00856.x

Age, gender, weight and height available for three subjects, other details undisclosed.

10B-concentrations were measured with prompt gamma ray spectroscopy (PGRS).

Fukuda H, Hiratsuka J, Honda C, et al. Boron neutron capture therapy of malignant melanoma using 10B-paraboronophenylalanine with special reference to evaluation of radiation dose and damage to the normal skin. Radiat Res. 1994;138(3):435-442. doi:10.2307/3578693

Age and gender of the Subjects are available, other details undisclosed.

10B concentration was measured by a chemical method, inductively coupled plasma atomic emission spectrometry (ICP-AES) or prompt y-ray analysis

Koivunoro H, Hippeläinen E, Auterinen I, et al. Biokinetic analysis of tissue boron (10B) concentrations of glioma patients treated with BNCT in Finland. Appl Radiat Isot. 2015;106:189-194. doi:10.1016/j.apradiso.2015.08.014

Demographics of the subjectnot described

inductively coupled plasma-atomic emission spectroscopy (ICP- AES)

The variation in analytical methods, along with the differences in dosing schedules and patient demographics as mentioned in response to Comment 1, contributes to the complexity of the data we analyzed. However, our approach using NONMEM software effectively accommodates these variations. By integrating the diverse analytical methodologies and their respective data into our PK model, we enhance the model's robustness and its applicability across different clinical settings. This integrative approach allows our predictive platform to be broadly relevant, offering valuable insights for BNCT treatment planning regardless of the specific analytical techniques used in individual studies.

In conclusion, while the variability in analytical techniques and dosing regimens presents challenges, it also strengthens the adaptability and applicability of our PK model in various BNCT scenarios.

**Response to Comment 3:

Thank you for your question regarding the variability parameters in the Table 2. Regarding the CV% of the parameters, the CV(%) in the table 2 is the interindividual variations (IIV) of the respective PK parameters such as K12, K21, and the additive (ε1) and proportional (ε2) errors are for residual error model. The IIV of the parameters are expressed as variance in log-normal distribution model, which is converted into CV(%), which does not have additive and proportional error. Likewise, K12 does not have additive and proportional error, and its IIV is expressed as CV(%) in the table 2.

Also, Please find attached revised manuscript updated per reviewer's comment

We hope these responses address your concerns and help improve our manuscript. Your feedback is invaluable to us.

Best regards,

Woo

Reviewer 3 Report

Comments and Suggestions for Authors

In this study, the authors present the results of a PK model developed for BNCT data. They analyzed literature data and developed a platform that can predict individual boron PK and the optimal time window for boron neutron capture therapy. 

They applied typical NLME approaches, and the novel aspect of the study is that they utilized BNCT C-t data. 

The methodology steps are clear and fully explained, while the so-derived PK model appears adequate, as shown from the validation findings.

Comments:

1. In Fig. 2, there appears to be a model underestimation for both cases. What measures did you take to avoid these poor model predictions?

2. It is stated that a user-friendly platform. It would be interesting for the readers to provide a snapshot of this platform to see its GUI.

3. Since, in this study, literature data were analyzed, and the BNCT data did not come from a trial performed in this study, I would suggest shortening the initial five paragraphs of the Discussion section.

Author Response

Dear Reviewer #3

Thank you for your feedback on our manuscript. We value your insights and have addressed each of your comments as follows:

Regarding the perceived underestimation in Figure 2: The figure aims to represent the population predicted concentration (red line) alongside the 95% prediction interval of individual distribution (pink shaded area), highlighting the consideration for interindividual variation. The key objective was to demonstrate that the observed data points (black dots) fall within this 95% prediction interval, thereby underscoring the model's accuracy across the population. This approach is consistent with our goal to illustrate the model's robustness in capturing individual kinetics within the observed population variation.

For comparison into prediction accuracy at the individual level, we direct attention to Figure 1B, which showcases the distribution of blood boron (10B) concentration measurements in relation to the baseline (y=x line), revealing a balanced dispersion around this reference. This visualization effectively conveys the model's precision in individual predictions.

The below figure, which is not included in the manuscript, shows the prediction accuracy in each individual in terms of B-10 concentration over time. The red line indicates the population mean and the dotted line indicates the individual prediction using the constructed pharmacokinetic model, which shows substantial high prediction.

The below figure was not included in the manuscript that it could be considered too much information and rather the Fig. 1B was included to show the prediction accuracy of each individual B-10 concentration in each sampling timepoints in overall. We have proposed not to include the below individual figure in the manuscript that it could be somewhat qualitative selective case review.

Rather, we have made it accessible as supplementary material, hoping this addresses your concerns and enhances the manuscript's comprehensiveness. Hope this response could meet your comment and expectation.

Thank you for the insightful comment on the snapshot of this platform. It would be informative to share readership regarding the use of the platform and we would share the below screenshot of the platfrom in the supplementary material to share the GUI.

The below screenshot shows the GUI of the platform which is based on Rshiny application when the raw data is submitted to the platfrom when the subject’s blood sample analysis is available. After the data was entered regarding the Infusion information and the also the blood B-10 concentration results, it goes through the calculation process and the results as shown in figure 4 in the manuscript is generated.

The accompanying screenshot displays the graphical user interface (GUI) using Rshiny application platform, designed for ease of use. Researchers or clinicians can input raw data, including infusion details and blood B-10 concentration measurements, from subjects' blood sample analyses directly into the platform. Upon data entry, the platform processes this information to generate the results depicted in Figure 4 of our manuscript. Including this GUI snapshot in the supplementary materials aims to provide readers with a clear view of the platform's functionality and practical application, facilitating its use in BNCT research and clinical settings.

Regarding shortening the initial five paragraphs, We have partially revised the paragraphs to shorten discussion section as below.

Original submitted draft of the first five paragraphs:

BNCT is a precise radiotherapy that utilizes boron-10’s ability to capture thermalized neutron, leading to a subsequent fission reaction that emits alpha and Li-7 particles, delivering high-LET (linear energy transfer) radiation to tissue at the cellular level. In BNCT, the precision of treatment planning and delivery is dependent upon two key factors: the amount of neutron and the boron concentration in each tissue. These parameters must be accurately determined or estimated in order to ensure effective treatment.

The accuracy of radiotherapy delivery is essential in achieving the desired balance between tumor control probability (TCP) and minimizing normal tissue complications probability (NTCP) as  demonstrated by dose response curves. In an effort to optimize radiotherapy outcomes and advance treatment techniques, it is imperative to consider the steepness of the TCP or NTCP curve as a key factor in determining the required accuracy of dose delivery. Any deviation from the planned dose, even in small amounts, can result in a decrease in TCP or an increase in NTCP, with the steepest curves being observed for normal tissue effects with a γ50 value of up to 6-7% per 1% change in dose, which highlights the importance of ensuring accurate dose delivery in clinical practice[16].

According to International Commission on Radiation Units & Measurements (ICRU) Report 24, the recommended accuracy for radiotherapy delivery should be 5% or less. Other studies, such as those conducted by Mijnheer et al and Brahme et al, have reported required accuracy levels by considering normal tissue complications and the impact on TCP, with values ranging from 7% to 3% relative standard deviation, respectively. These findings underscore the need for ongoing efforts to improve accuracy in radiotherapy delivery and maintain the balance between maximizing tumor control and minimizing tissue complications.

In BNCT, multiple quality assurance items are performed to ensure accurate radiation dose, including verification of dose calculation algorithms, validation of treatment planning systems, and examination of the physical characteristics of the neutron beam of the therapeutic neutron device. Even with the implementation of various quality assurance measures of physical properties of neutron and devices, the challenge of accurately estimating and monitoring the changes in blood boron concentration of boron pharmaceuticals in vivo remains as an outstanding issue of uncertainty. Previous attempts to address this challenge have involved using linear estimation or a combination of 2-compartment and biphasic estimation with spreadsheet tools with high accuracy.

In a clinical study conducted at Brookhaven National Laboratory in the late 1990s[17], the average blood concentration was calculated by linear extrapolation of blood concentrations just before, during, and after neutron irradiation in the nuclear reactor, respectively. From 1996 to 1999, Harvard Institute of Technology and MIT In a clinical study jointly conducted[18], it was revealed that the blood concentration was modeled by referring to the fact that the blood boron concentration showed a bi-exponential washout in the existing literature, and prediction and post-testing were performed based on this as a reference. In Finland, a 2-compartmental model and a bi-exponential model were constructed using the patient blood concentration of the study conducted at the Brookhaven National Laboratory and predicted based on these[19]. In the sponsor-led clinical trial of Japan’s Stellar Pharma and Sumitomo Heavy Industries, which received product approval as a BNCT for recurrent head and neck cancer, and the sponsor-led clinical trial conducted in glioblastoma, the time corresponding to a specific dose based on the blood concentration immediately before neutron irradiation. A separate PK model for the prediction of blood boron concentration during neutron irradiation is not disclosed.

————————

Summarized draft of the first five paragraphs:

BNCT employs boron-10 for high-LET radiation delivery to cancer cells, requiring precise neutron and boron tissue concentration measurements. Achieving accuracy is vital, with dose-response studies indicating significant impacts on Tumor control probability (TCP) and normal tissue complications (NTCP) from small dose deviations. Highlighting the necessity of stringent accuracy, dose effects in normal tissues show a γ50 value of up to 6-7% per 1% dose change, advocating for dose delivery precision within a 5% margin as recommended by ICRU Report 24.

Our study introduces a NONMEM-based prediction model for blood boron levels during BNCT, aiming to enhance treatment precision. This effort is aligned with the historical pursuit of accuracy in BNCT, as evidenced by various clinical studies. At Brookhaven National Laboratory in the late 1990s, average blood concentrations were linearly extrapolated around neutron irradiation times. Joint studies by Harvard Institute of Technology and MIT from 1996 to 1999 utilized bi-exponential washout models, reflecting the nuanced behavior of blood boron concentration. Finnish research applied a 2-compartmental model and a bi-exponential model, leveraging Brookhaven's patient data for predictive accuracy. In Japan, Stellar Pharma and Sumitomo Heavy Industries' clinical trials for BNCT in head, neck cancer, and glioblastoma did not disclose a specific PK model for irradiation-time blood boron concentration prediction. These historical approaches underscore the complexities and challenges in accurately predicting boron concentration, a critical factor our study seeks to address with improved precision and reproducibility

--------------------------

Thank you for your thoughtful recommendations. In our original draft, we presented a detailed discussion on the importance of achieving dose precision in BNCT, which we believe is essential to thoroughly justify our focused approach on estimating B-10 concentration accuracy. This detailed explanation is pivotal for readers to understand the significant challenges and uncertainties associated with BNCT dosing, and the potential clinical implications of these uncertainties.

We value the editorial feedback and acknowledge that there may be different perspectives on the level of detail necessary. If it is the consensus that a more streamlined discussion would benefit the readership, we are open to carefully revising our manuscript accordingly. We would appreciate any additional guidance you may have to ensure that our revisions meet the journal's standards.

Please review the attached manuscript updated according to your comments. We have retained the initial five paragraphs in their original form to fully address the importance of B-10 concentration accuracy in BNCT and the resulting clinical implications. But still, If you strongly feels that revisions are needed, we are willing to amend these paragraphs cautiously, ensuring the core message of our study remains clear.

Also, Please find attached updated manuscript revised per reviewer's comments for reference.

We hope these responses address your concerns and help improve our manuscript. Your feedback is invaluable to us.

Best regards,

Woo

Round 2

Reviewer 1 Report

Comments and Suggestions for Authors

Thank you for your sincere response to my comment.

Reviewer 3 Report

Comments and Suggestions for Authors

Thank you for addressing my comments. I have no further suggestions to make.